

# Stratospheric observations of noctilucent clouds: a new approach in studying large-scale mesospheric dynamics

Peter Dalin[1,2,*], Nikolay Pertsev[3], Vladimir Perminov[3], Denis Efremov[4,5], Vitaly Romejko[6]

[1] Swedish Institute of Space Physics, Box 812, SE-981 28 Kiruna, Sweden

[2] Space Research Institute, RAS, Profsouznaya st. 84/32, Moscow, 117997, Russia

[3] A.M. Obukhov Institute of Atmospheric Physics, RAS, Pyzhevskiy per. 3, Moscow, 119017, Russia

[4] Aerospace laboratory õStratonauticaö, Moscow, Russia

[5] Faculty of Cosmic Research, M.V. Lomonosov Moscow State University, GSP-1, Leninskie Gory, Moscow, 119991, Russia

[6] The Moscow Association for NLC Research, Kosygina st. 17, Moscow, 119334, Russia

*Corresponding author at: Swedish Institute of Space Physics, Box 812, SE-981 28 Kiruna, Sweden. Fax: +46 980 79050. E-mail address: pdalin@irf.se (P. Dalin).

**Abstract.**

The experimental campaign Stratospheric Observations of Noctilucent Clouds (SONC) was conducted on the night of 5-6 July 2018 with the aim of photographing noctilucent clouds (NLC) and studying their large-scale spatial dynamics at scales of 100ó1450 km. An automated high-resolution camera (equipped with a wide-angle lens) was lifted by a stratospheric sounding balloon to 20.4 km altitude above the Moscow region in Russia (~56°N; 41°E), taking several hundreds of NLC images during the flight that lasted 1.7 hours. The combination of a high-resolution camera and large geographic coverage (~1500 km) have provided a unique technique of NLC observations from the stratosphere, which is impossible to currently achieve either from the ground or space. We have estimated that a horizontal extension of the NLC field as seen from the balloon was about 1450 x 750 km whereas it was about 800 x 550 km as seen from the ground. The NLC field was located in a cold area of the mesopause (136-146 K), which is confirmed by satellite measurements. The southmost edge of the NLC field was modulated by partial ice voids of 150-250 km in diameter. A medium-scale gravity wave had a wavelength of 49.4±2.2 km with vertical amplitude of 1.9±0.1 km. The final state of the NLC evolution was represented by thin parallel gravity wave stripes. Balloon-borne observations provide new horizons in studies of NLC at various distances from metres to thousands of km.



34

*Keywords*: noctilucent clouds, mesospheric dynamics, balloon-borne stratospheric observations, atmospheric gravity waves

37

## 1 Introduction

Night-shining clouds or noctilucent clouds (NLC) are the highest clouds in the Earth's atmosphere observed at the summer mesopause between 80 and 90 km. NLC can be readily seen from mid- and subpolar latitudes of both hemispheres. NLC are composed of water-ice particles of 30–100 nm in radius that scatter sunlight and thus NLCs are observed against the dark twilight arc from May until September in the Northern Hemisphere and from November to February in the Southern Hemisphere (Bronshten and Grishin, 1970; Gadsden and Schröder, 1989; Liu et al., 2016). NLC are also observed from space and in this case they are usually called Polar Mesospheric Clouds (PMC) (Thomas, 1984).

NLC are almost always represented by a wave surface having a complex interplay between small-scale turbulence processes of 10-1000 metres, atmospheric gravity waves (GW) with wavelengths of 10-1000 km, planetary waves, solar thermal tides and lunar gravitational tides of about 10000 km (Witt, 1962; Fritts et al., 1993; Rapp et al., 2002; Kirkwood and Stebel, 2003; Chandran et al., 2009; Dalin et al., 2010; Fiedler et al., 2011; Taylor et al., 2011; Pertsev et al., 2015). Sometimes, distinguished non-linear mesospheric phenomena like mesospheric walls or fronts appear at the mesopause which clearly separate two volumes of the mesopause having cold and warm air masses with temperature difference of 20-25 K across a few km (Dubietis et al., 2011; Dalin et al., 2013).

NLC/PMC are systematically observed and studied from the ground (optical imagers, lidars), as well as from space (AIM, Odin, SBUV instruments) (e.g., Karlsson and Gumbel, 2005; Dalin et al., 2008; Bailey et al., 2009; Fiedler et al., 2011; DeLand and Thomas, 2015); there are also irregular (campaign-based) NLC observations conducted by using sounding rockets and aircraft (Zadorozhny et al., 1993; Gumbel and Witt, 2001; Reimuller et al., 2011). These techniques have advantages and disadvantages. In particular, ground-based measurements provide a high horizontal resolution of ~20 m and high temporal resolution of seconds (optical imagers) (Dalin et al., 2010; Baumgarten and Fritts, 2014) and high vertical resolution of 50-150 metres using lidars (Baumgarten et al., 2009) but are limited to tropospheric weather conditions and restricted to a certain small region on the Earth's surface.




Satellite measurements, on the other hand, provide global PMC coverage but have low spatial
horizontal resolution (~5 km) as well as large spatial gaps of several hundreds of km between
adjacent orbits at middle and subpolar latitudes. Thus, there is no perfect technique to observe
and study NLC/PMC so far. There is an obvious methodological gap in these techniques,
resulting in a gap of stratospheric altitudes (20-40 km), which are potentially available for
comprehensive studies of NLC/PMC. This gap is due to lack of systematic stratospheric
balloon-borne experiments aiming at NLC/PMC observations. So far, there has been
conducted a single published experiment from a stratospheric balloon providing PMC
observations over Antarctica between 29 December 2012 and 9 January 2013 (Miller et al.,
2015). The E and B Experiment (EBEX) was dedicated to another research field concerning
polarization in the cosmic microwave background (Reichborn-Kjennerud et al., 2010). At the
same time, two star cameras of the EBEX experiment, having a narrow field of view of 4.1° x
2.7°, were able to register fine structures of PMC and turbulence dynamics, ranging from
several km down to 10 m. Another balloon-borne experiment (PMC-Turbo) was conducted
between 8 and 14 July 2018 over Sweden-Greenland-Canada territories in order to capture
NLC with seven optical cameras and lidar (Fritts et al., 2019). The PMC-Turbo experiment
was launched about 2.5 days after the experiment described in the present paper.

In this paper, we report on scientific results of a new balloon-borne experiment dedicated

to studies of NLC large-scale dynamics at horizontal scales of more than 100 km (Dalin et al.,
2019). Such experiment, conducted for the first time, opens new horizons for studies of large-
scale dynamical features in combination with a high spatial resolution at the summer
mesopause, currently unachievable for other techniques like ground-based and space
measurements.

**2 Technique and method**

The Stratospheric Observations of Noctilucent Clouds (SONC) experiment is a special

balloon-borne experiment dedicated to studies of large-scale dynamical features in NLC. A
high resolution high sensitive camera (Sony Alpha A7S), having a full frame 35 mm 12
megapixel sensor (4240 x 2832 pixels) and equipped with a wide-angle lens (field of view,
FoV, is 109.7° x 81.6°), has been installed on a meteorological sounding balloon. This
combination of a high resolution sensor and wide FoV yields spatial horizontal resolutions of
~30 m and ~3000 m, when looking at 83 km from 20 km at elevation angles of 90° and 0°,
respectively. The horizontal coverage of a mesopause layer is over 2000 km, when viewing
along the horizon at low elevation angles. The balloon was launched from the Moscow





region, Russia (~56°N; 41°E), on the night of 5-6 July 2018. Since a gondola payload is
constantly rotating and shaking during its flight, the NLC camera was installed on a special
stabilized platform. The 3-axis motorized gimbal stabilized platform (Fig. 1) was designed
and build by the Aerospace laboratory õStratonauticaö (http://stratonautica.ru), which has a
wide experience in building such platforms and launching sounding balloons. The platform
was designed to rotate in a 60° step in the azimuth angle in order to capture the whole
hemisphere (360°) since NLC can appear in any direction as observed from mid-latitudes,
including the southern part of the sky (Hultgren et al., 2011; Suzuki et al., 2016). The NLC
camera took images every 6 s during the whole flight, obtained several thousands of images
and several hundreds of images capturing NLC. Besides, automatic exposure bracketing was
used to take four images in sequence with different exposures, allowing us to register various
NLC brightness from very bright to very faint as well as faint stars, which are important
information for the photogrammetric technique and georeference procedure of the images.

The balloon was launched at 21:34 UT on 5 July 2018 and the total flight duration was

about 1.7 hours. The ascent speed was around 5 m/s and the balloon reached its maximum
altitude of 20.4 km where it burst; then the payload descended with a parachute and the
payload was successfully recovered. A GPS receiver was installed onboard in order to obtain
information on the balloon trajectory. The flight characteristics of the SONC balloon are
shown in Fig. 2.

A ground-based support consisting of three automated NLC cameras was established in

the Moscow region in order to launch the balloon at the time of NLC appearance. Also, a
number of amateur observers significantly contributed to the NLC observational programme
before and during the flight. A launch window was preliminary chosen at the beginning of
July based on long-term statistics of NLC observations conducted in the Moscow region since
1962 to present time. This statistics demonstrate that NLC appear at the beginning of July
with about 65% occurrence probability on a clear night.

**3 The observation**

During the flight, the balloon-borne camera captured an extended NLC field with a

number of interesting features discussed in section 4. One can note the following general
characteristics of the NLC display:

a) NLC were observed between 20:30 and 23:15 UT on 5 July 2018.



b) NLC were located between 82.6 and 85.1 km. The NLC height was estimated by using
synchronously taken images obtained from two ground-based cameras located in the Moscow
region.
c) NLC field extended along the horizon from NW to NE at low elevation angles from 65°
to +11° as seen from the balloon.
d) NLC were modulated by atmospheric gravity waves of various scales having horizontal
wavelength from 9 km to 50 km.
e) NLC were traveling in a rather unusual direction from the south to north at the observed
mean speed of ~43 m/s.
f) NLC were fading during the balloon ascent and they got very faint and less structured at
the maximum balloon altitude of 20.4 km. The brightest and well-developed NLC were
observed when the SONC balloon was between 6 and 13 km, that is why we analyze the most
profound features of NLC images obtained at this height range.
Each analyzed NLC image was georeferenced using horizontal coordinates of referenced
stars (at least 15 stars are needed). The technique of the NLC georeference, triangulation
height estimation and error analysis can be found in Dalin et al. (2004, 2015).

**4 Results and discussion**
The projection of the NLC field on the surface along with the temperature map obtained
with the Aura/MLS spectrometer is shown in Fig. 3. The description on the MLS temperature
product and its validation can be found in Froidevaux et al. (2006) and Schwartz et al. (2008).
One can see that the NLC field extended mostly from the west to east along an area filled with
low temperatures of 136-146 K, and the NLC were located north of 58°N due to rapidly
increasing temperature with decreasing latitude. That is why the NLC were observed at low
elevation angles (far to the north as seen from the Moscow region) on this particular night.
Detailed analysis of five consecutive in time balloon-borne images (Figs. 4 and 5) has
revealed the following features of the NLC display:
a)  The horizontal extent of the NLC field from the western to eastern observable border
was about 1450 km, and from the northern to southern border of about 750 km. Such
distances are impossible to observe from the ground due to the Earth's curvature and limited
area of the twilight arch. The central part of the NLC field, having extension of about 850 x
550 km, was seen from the ground but the western and eastern wings of the field as well as
the northern edge were located below the local ground horizon, making it impossible to
observe them. Thus, balloon-borne NLC observations have an obvious great advantage over



ground-based observations in terms of larger geographic coverage which is comparable to
PMC observations made from space.
b) The southmost edge of the NLC field was modulated by partial circles (something like
ice voids but with open southern border), which is shown by the red curves in Figures 4 and 5.
The diameters of these partial ice voids are estimated to be in the range of 150-250 km. The
mechanism of the formation of ice voids in NLC/PMC is not clear now, and it is an ongoing
topic in atmospheric physics. One can mention three main mechanisms which are currently
discussing in the literature. Trubnikov and Skuratova (1967) addressed a theory of cellular
convection and demonstrated its principal possibility in the summer mesosphere in relation to
NLC occurrences. The authors estimated convective cells to be in the range of 90-250 km in
radius, that agrees well with sizes of partial ice voids obtained in the present study. However,
there should be fulfill the main criterion for the convection to be developed, namely, the
height gradient of the potential temperature should have negative values. We have carefully
estimated the potential temperature gradient or the static stability (based on Aura/MLS
temperature measurements) in the analyzed area and could not find any signatures of its
negative values in the mesosphere and mesopause region. It means that in this particular case
cellular convection could not be responsible for the observed partial ice voids in the NLC.
However, satellite measurements can easily miss a negative static stability at local scales
due to poor horizontal resolution and local ice voids may be generated by a gravity wave
breaking. Rusch et al. (2009) have hypothesized that ice voids could be caused by heating due
to the passage of warm crests of a gravity wave. It is possible in the present case. However,
we could not find any significant displacement of the partial ice voids (their boundaries)
relative to the NLC field, i.e., the partial ice voids traveled with the same speed and direction
as the entire NLC field did (~43 m/s from the south to north). One would expect an intrinsic
phase speed and intrinsic direction of the movement of the partial ice voids if they were
generated by a large-scale gravity wave of a wavelength of several hundreds of km. Thus, it is
difficult to prove this hypothesis of the influence of a large-scale gravity wave on the
formation of the observed partial ice voids.
Thurairajah et al. (2013b) have proposed another mechanism related to a shock wave
generated by a meteorite, which expands and cools the air that in turn leads to the formation
of large ice particles which fall out of an NLC field (analogously to hole-punch clouds due to
the passage of an aircraft). However, we observe large-scale partial ice voids (150-250 km) in
a broad area of the mesopause over 1000 km. It was hardly possible that any big meteorite



could produce such large holes in such broad area, and we did not observe any meteor motion
in our ground-based and balloon images.
Megner et al. (2018) have recently presented an interesting case study of a quasi-
stationary ice void in NLC which did not follow the general wind, suggesting that it was
formed by a localized warming at the summer mesopause. This is not the case in our case
study, in which we have observed partial ice voids moving at the general wind speed in the
same direction along with the entire NLC field.
In the present case study, the partial ice voids had irregular shape and sizes ranging from
150 to 250 km. Also, these partial voids moved along the wind, having the same speed and
direction. Thus, it is difficult to connect these partial voids with regular wave disturbances. At
the same time, as shown in Fig. 3, the southmost border of the NLC field was confined to the
warm air mass located at sub-polar latitudes of ~58°N and lower. The mesopause temperature
at this border was equal to ~147 K at 86 km altitude. The MLS data cannot reproduce the
exact shape of this border due to low horizontal resolution (~15°) and temporal resolution of
~1.5 h. However, it is well known that tropospheric frontal systems have a meandering shape,
sometimes with intrusions of warm and cold air masses as in case of the formation of a frontal
wave cyclone (Ahrens, 1993; Stull, 2000). In our case the warm front at the mesopause and
the NLC partial ice voids resemble a tropospheric frontal wave, in which there are intrusions
of warm air masses, moving from midlatitudes, into the cold air mass located at sub-polar and
polar latitudes (see Fig. 6). Therefore, we consider that the most probable source of these
partial ice voids observed in the NLC in this particular case is the intrusion of warm air
masses into the cold air mass with the NLC field, sublimating ice particles. A similar
conclusion was proposed by Thurairajah et al. (2013a) who have analyzed a large ice void
observed in PMC (using AIM/CIPS satellite images) and have concluded that õí *warmer*
*temperatures (warmer than the frost point temperature of ~144 K) at the location of the void*
*may be related to increased tidal activity and transport of warm air from low latitudes*.ö Also,
Bailey et al. (2009) and Thurairajah et al. (2013b) have demonstrated that southmost borders
of PMC can be highly modulated by partial ice voids of several hundreds of km in diameter,
and the authors have found the structural similarity between PMC images and those seen in
tropospheric clouds.
c) Clear vertical modulation of the NLC layer is shown with the red arrow in Fig. 7. This
is a unique view on a particular gravity wave seen at the local horizon of the balloon; that is
why this wave modulation is viewed almost at the right angle to the line-of-sight. This
geometry allows observing a thin layer of NLC modulated in altitude by propagating gravity



waves of small and medium scales. Such geometry is almost impossible to obtain from the
ground since NLC seen at the very horizon are usually masked by topography, tropospheric
clouds and, most importantly, by tropospheric aerosols, which are constantly present and
significantly absorb NLC brightness when looking at the very horizon. We have carefully
estimated parameters of this particular wave: its horizontal wavelength was equal to 49.4±2.2
km and its vertical amplitude was 1.9±0.1 km between the crest and trough. In this
calculation, the angle of 13.3° between the camera image plane and vertical plane at the NLC
altitude was taken into account. Also note that since NLC are clearly seen both in the crest
and trough of the wave (ice particles did not completely sublimated in the wave trough), we
have estimated the wave amplitude both in the wave crest and trough. The amplitude
estimations are the same in the wave trough and crest (within the given uncertainty). All this
makes us confident in the estimation of the vertical amplitude of this particular wave. This is
the most precise estimation of the amplitude of a gravity wave at the mesopause by using
NLC observations (Witt, 1962; Haurwitz and Fogle, 1969; Bronshten and Grishin, 1970;
Demissie et al., 2014). Since wave amplitude represents kinetic wave energy, this is an
important source of information for estimating the wave energy budget at the upper
atmosphere, and also can be used for future model studies to estimate a wave source in the
lower atmosphere (Fritts and Alexander, 2003; Demissie et al., 2014).
d) Small-scale billow-type gravity waves were estimated to have horizontal wavelengths
of 8-11 km (Fig. 7). Such small-scale gravity waves are well-known to be observed in NLC
layers (Witt, 1962; Dalin et al., 2010; Pautet et al., 2011; Baumgarten and Fritts, 2014;
Demissie et al., 2014), but we demonstrate this result in order to emphasize the ability to
resolve small-scale NLC structures by using a large FoV camera, having a high resolution
sensor, onboard a sounding balloon.
e) Figure 8 illustrates an NLC image taken from altitude of 20.3 km which is very close to
the maximum reached altitude of 20.4 km. The NLC were rather faint by that time that is in
line with an idea of the intrusion of warm air masses from mid- to subpolar latitudes. These
large-scale warm air masses led to rapid sublimation of ice particles at large scales of about
1500 km. At the same time, on can see a very interesting feature to be considered. There were
several thin parallel gravity wave bands (stripes) with lengths of 50-200 km and widths of ~
3-5 km in cross-section. The reasons of seeing such thin stripes are as follows: (a) The SONC
balloon was in the stratosphere, i.e., above the troposphere in which optically strong air
turbulence is constantly present (b) The exposure time of this image was very short of 1/125 s.
All these made the image free from blurring (as minimum blurring as possible for moving





NLC and balloon motion). This image demonstrates a final stage of the NLC evolution (NLC
disappeared in 20 min since the image was taken), and these thin stripes might represent a
final morphological state of the NLC evolution. Further balloon-borne NLC observations of
very faint NLC are required to confirm this consideration.

**5 Conclusions**
The combination of high resolution images (~30 m) and large geographic coverage (over
1500 km) is a unique property intrinsic to stratospheric balloon-borne NLC observations,
which is impossible to achieve either from the ground or space. In general, a balloon-borne
NLC observation provides us with the following new opportunities in case of a long duration
flight of several days:
a) NLC imaginary can be obtained for 24 hours a day and during several days due to very
little Rayleigh atmospheric scattering in the visible subrange of the spectrum above 20 km
(Hughes, 1964);
b) Quantitative information on a wide range of waves (gravity and planetary waves, solar
tides), propagating through the summer mesopause can be obtained;
c) Neutral wind velocity at the mesopause and large-scale trajectory of NLC fields over 1500
km can be measured;
d) Quantitative information on long mesospheric fronts, solitons and other non-linear
processes can be obtained;
e) Quantitative information on small-scale turbulent structures (down to 1 m) can be
obtained in case of using a narrow field of view lens.
f) High resolution vertical NLC structure (wave modulation, double layers) can be retrieved
by observing NLC at the very horizon. Absence of any terrain obstacles and tropospheric
aerosol loading makes such stratospheric NLC observations unique.
g) Absence of optically strong tropospheric turbulence makes NLC images free from
atmospheric blurring that in turn results in well-defined fine structures of gravity waves
and turbulence in the mesopause region.

In the present study, we have estimated the following characteristics of the NLC field:
a) The horizontal extent of the NLC field as seen from the SONC balloon was about

1450 x 750 km whereas it was about 800 x 550 km as seen from the ground. This

emphasizes the great advantage of making large-scale balloon-borne observations over





medium-scale ground-based ones.
b) NLC field was traveling from the south to north at a mean velocity of 43 m/s;
c) The southmost edge of the NLC field was modulated by partial ice voids of 150-250
303       km in diameter, which were like generated by the intrusion of warm air masses
moving from mid- to sub-polar latitudes. The mesopause temperature at this edge was
equal to ~147 K, i.e., it was a threshold temperature separating the mesopause region
filled with NLC from the warm area without NLC.

d) A medium-scale wave had a wavelength of 49.4±2.2 km and vertical amplitude of
1.9±0.1 km. This is the most precise estimation of a gravity wave amplitude ever
made.

e) Small-scale billow-type gravity waves had wavelengths of 8-11 km.
f) The final morphology state of the NLC evolution was represented by thin parallel
gravity wave stripes with lengths of 50-200 km and widths of ~3-5 km.


*Data availability*. The reader can access the SONC experiment images and balloon GPS
coordinates, used in the paper, via publically available project ftp server at the Swedish
Institute of Space Physics: ftp://ftp.irf.se/outgoing/pdalin/NLC/SONC_experiment/

*Author contributions*. PD wrote the paper, made calculations and plotted the figures. NP and
VP read and made suggestions appropriated for the paper. DE provided the raw balloon-borne
images and balloon GPS coordinates. VR contributed to the image processing. All the authors
read and commented regarding the work and agreed with the content and submission of this
paper.

*Competing interests*. The authors declare that they have no conflict of interest.

*Acknowledgments*. The authors are grateful to Nikolay Gusev, Andrey Reshetnikov, Alexander
Dalin for their support of ground-based NLC observations during the SONC experiment. The
Aura/MLS data version 2.2 were obtained from the NASA Goddard Space Flight Center Data
and Information Services Center: https://mirador.gsfc.nasa.gov.



*Financial support*. The work was partly supported by the Russian Foundation for Basic
Research under project 15-05-04975a.

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





**Figure captions:**

**Figure 1**. The 3-axis motorized gimbal stabilized platform, holding the NLC camera, designed and build by the Aerospace laboratory õStratonauticaö. Photo by Denis Efremov.

**Figure. 2**. (Left) the altitude of the SONC balloon as a function of time flight. (Right) the vertical-horizontal trajectories of the SONC balloon: the red line is the upleg and the black line is the downleg trajectories.

**Figure 3**. The temperature map at the mesopause (86.1 km) as measured by the Aura/MLS spectrometer on 5 July 2018. Nighttime measurements around the globe have been selected to produce the map. Upon the temperature map, the outer borders of the NLC field are overplotted: the red line is as seen from the SONC balloon, the black line is as seen from the ground at the launch. The black dots mark the position of the balloon at 7.8 km at the ground and ground-based observers.

**Figure 4**. The NLC field as observed from the SONC balloon at 4092 m, 4947 m, 7836 m, 9077 m and 13928 m above the ground at 21:46 UT, 21:49 UT, 21:57 UT, 22:01 UT, 22:20 UT on 5 July 2018. The red curves indicate large areas free from NLC particles (partial ice voids).

**Figure 5**. Projection of the NLC fields (shown in Figure 4) as observed from the SONC balloon on the surface. The red curves indicate large areas free from NLC particles (partial ice voids).

**Figure 6.** A schematic representation of the intrusion of warm air masses from mid- to sub-polar latitudes, forming partial ice voids in the observed NLC. A general concept of this scheme is analogous to the formation of a wave cyclone in the troposphere (see Figs. 8.18 and 8.19 in Ahrens, 1993).

**Figure 7.** The SONC balloon image taken at 6222 m above the ground at 21:49 UT on 5 July 2018. The red arrow marks the vertical modulation of the NLC layer by a gravity wave of medium scale. The green arrow indicates small-scale billow-type gravity waves.

**Figure 8**. The SONC balloon image taken at 20.3 km above the ground at 22:48 UT on 5 July 2018 represents the final stage of NLC evolution on that night.





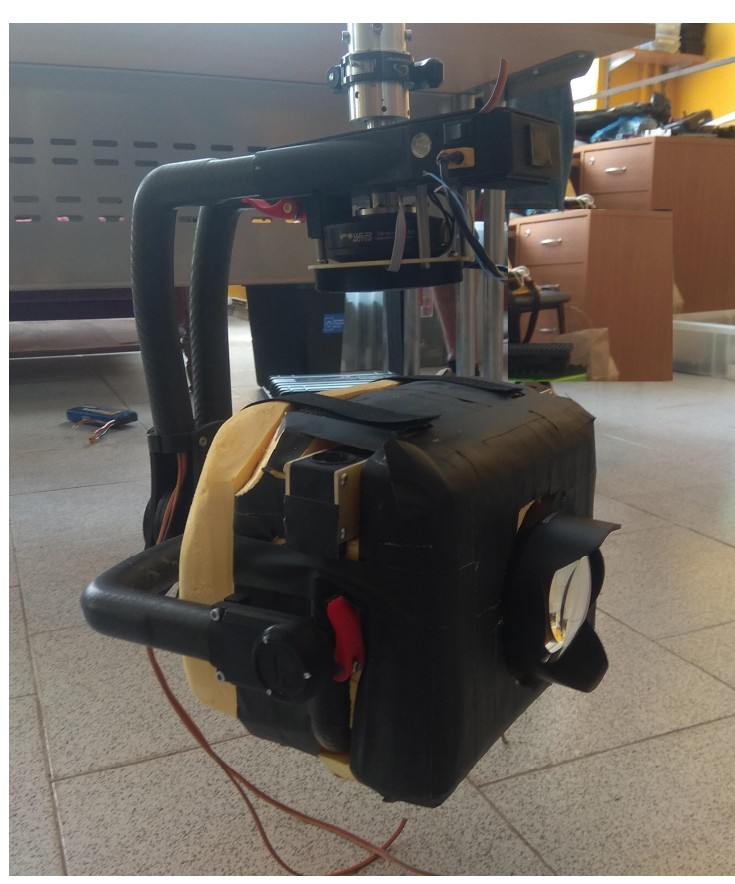

507

**Figure 1**. The 3-axis motorized gimbal stabilized platform, holding the NLC camera,

designed and build by the Aerospace laboratory õStratonauticaö. Photo by Denis Efremov.





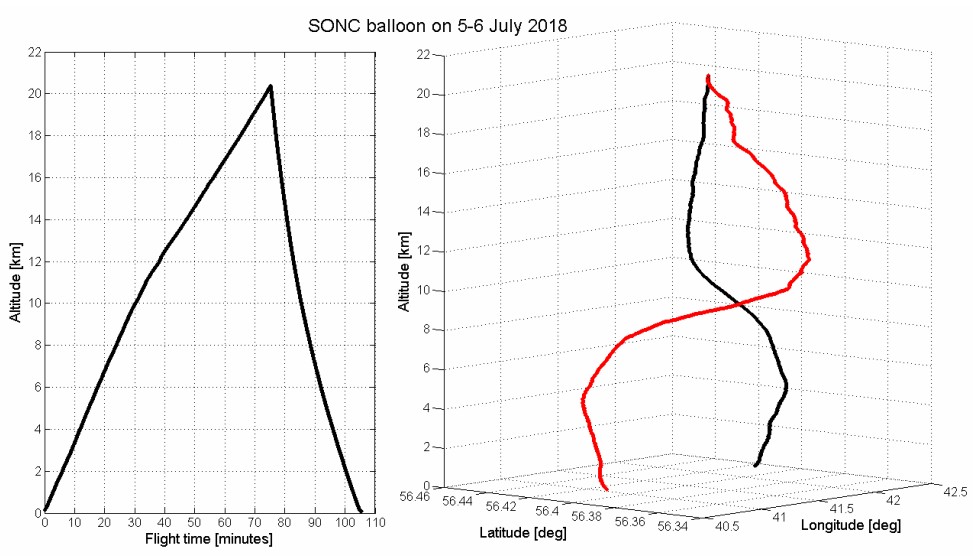

**Figure. 2**. (Left) the altitude of the SONC balloon as a function of time flight. (Right) the vertical-horizontal trajectories of the SONC balloon: the red line is the upleg and the black line is the downleg trajectories.



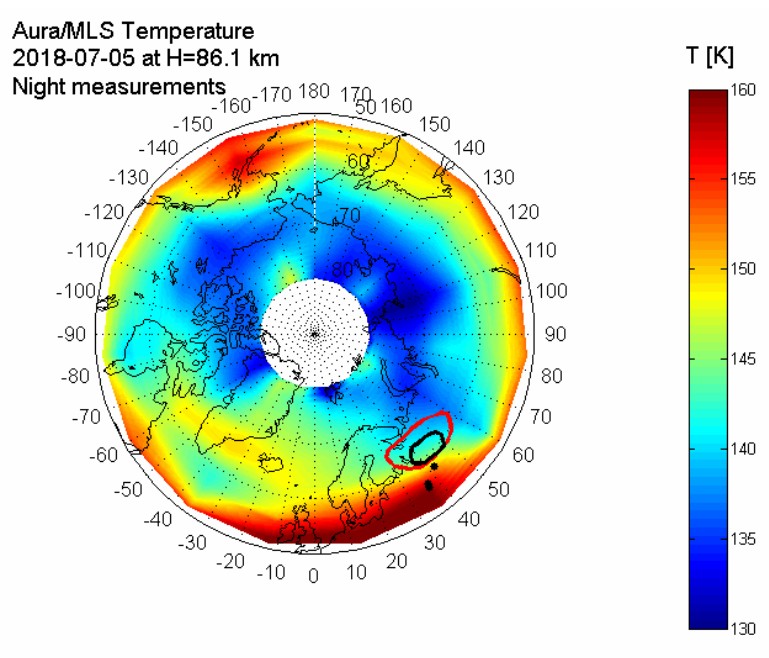

514

**Figure 3**. The temperature map at the mesopause (86.1 km) as measured by the Aura/MLS

spectrometer on 5 July 2018. Nighttime measurements around the globe have been selected to

produce the map. Upon the temperature map, the outer borders of the NLC field are

overplotted: the red line is as seen from the SONC balloon, the black line is as seen from the

ground at the launch. The black dots mark the position of the balloon at 7.8 km at the ground

and ground-based observers.





NLC on 5-6 July 2018,  H=4092 m

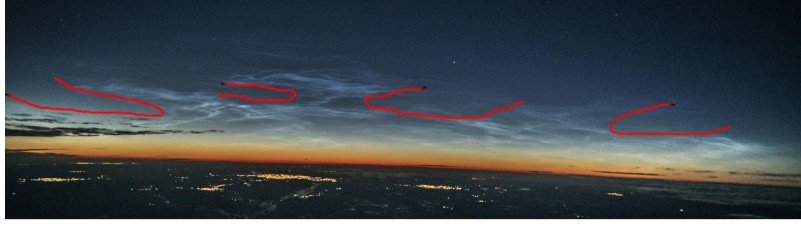

H=4947 m

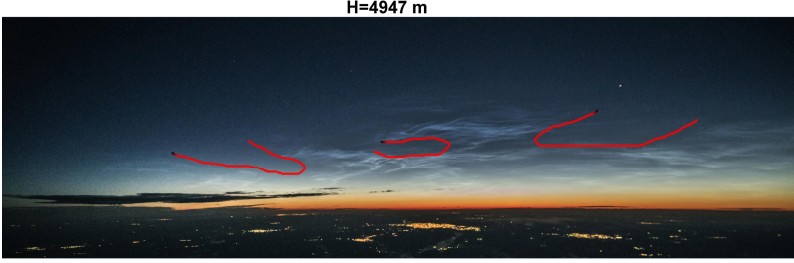

H=7836 m

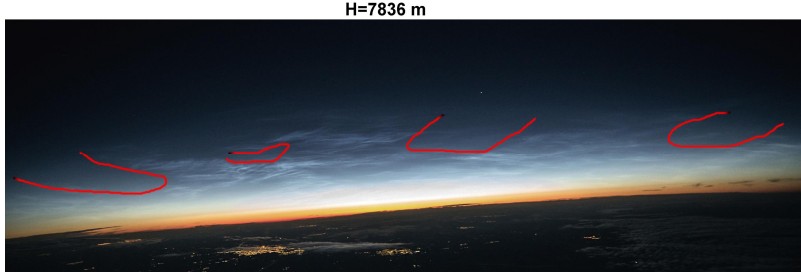

H=9077 m

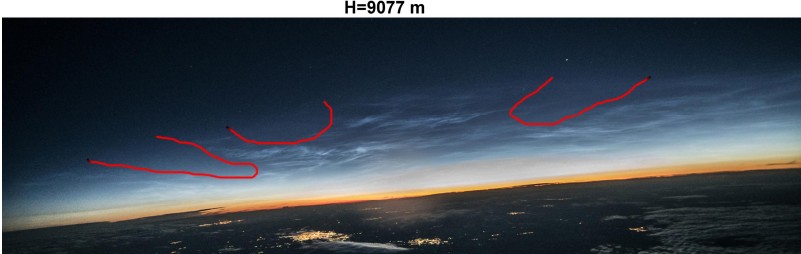

H=13928 m

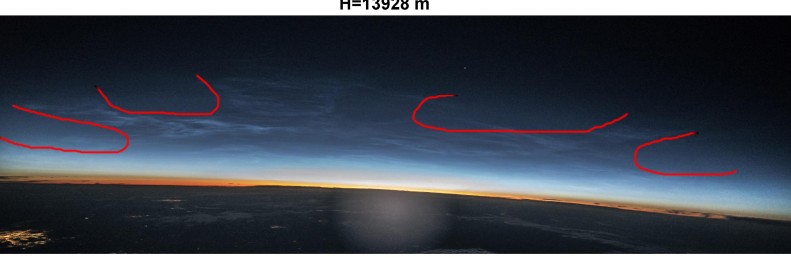

521

**Figure 4**. The NLC field as observed from the SONC balloon at 4092 m, 4947 m, 7836 m, 9077 m and 13928 m above the ground at 21:46 UT, 21:49 UT, 21:57 UT, 22:01 UT, 22:20 UT on 5 July 2018. The red curves indicate large areas free from NLC particles (partial ice voids).





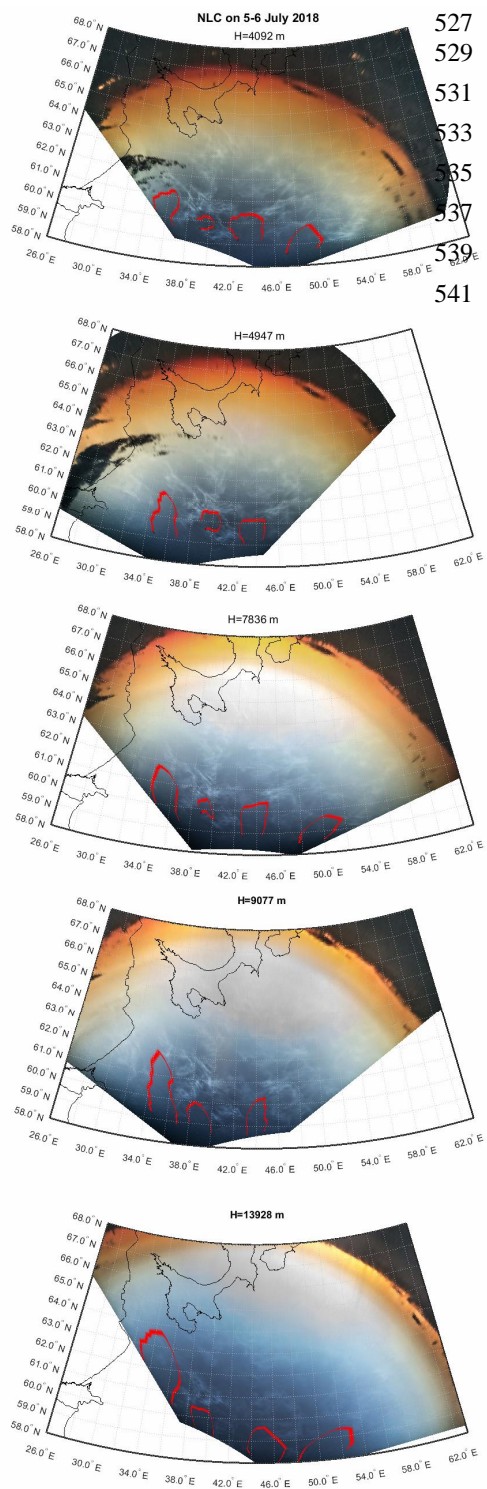


**Figure 5**. Projection of the NLC fields (shown in Figure 4) as observed from the SONC balloon on the surface. The red curves indicate large areas free from NLC particles (partial ice voids).



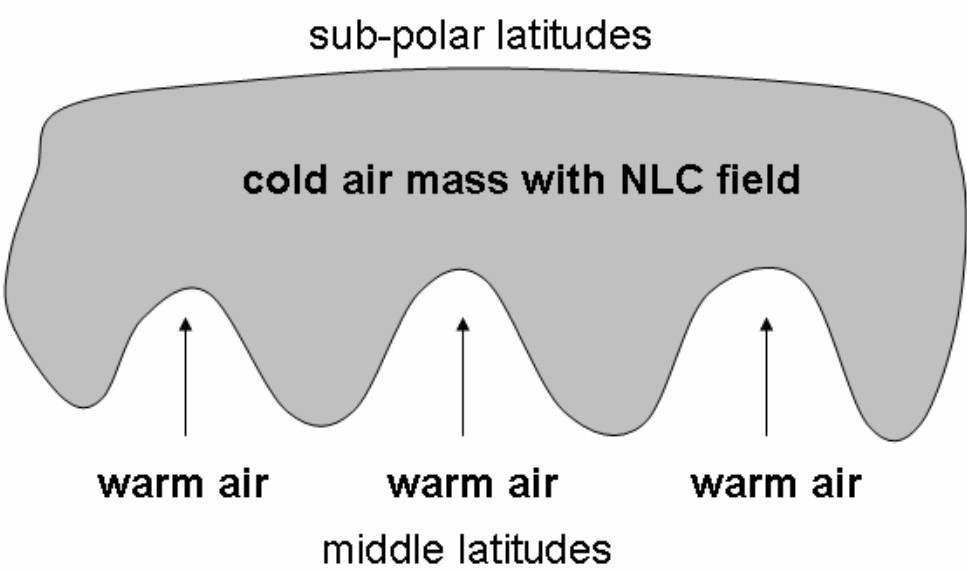


**Figure 6.** A schematic representation of the intrusion of warm air masses from mid- to sub-

polar latitudes, forming partial ice voids in the observed NLC. A general concept of this

scheme is analogous to the formation of a wave cyclone in the troposphere (see Figs. 8.18 and

8.19 in Ahrens, 1993).



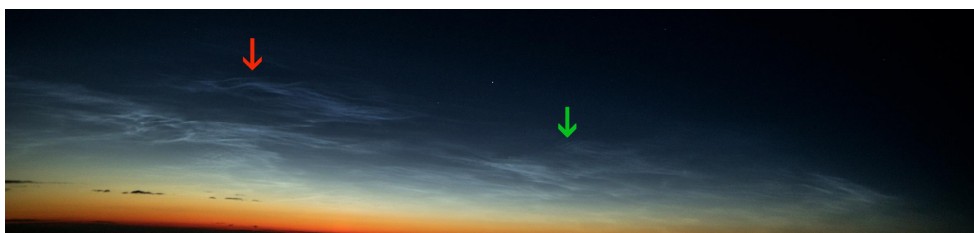

547

548

**Figure 7.** The SONC balloon image taken at 6222 m above the ground at 21:49 UT on 5 July

2018. The red arrow marks the vertical modulation of the NLC layer by a gravity wave of

medium scale. The green arrow indicates small-scale billow-type gravity waves.



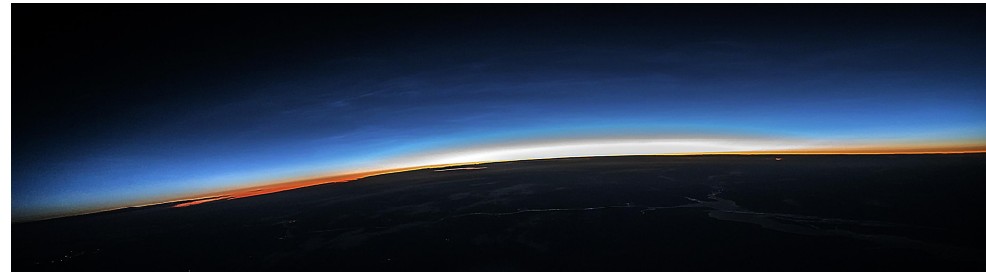

**Figure 8**. The SONC balloon image taken at 20.3 km above the ground at 22:48 UT on 5 July
2018 represents the final stage of NLC evolution on that night.