# Peer review of "Stratospheric observations of noctilucent clouds: a new approach in"

_Annales Geophysicae, 2019_

## Referee Comment (RC1) · Anonymous Referee #1 · 15 Sep 2019

This paper reports the stratospheric observations of noctilucent clouds from a balloon on 5-6 July 2018 near Moscow. Several hundreds of NLC images were taken for over one hour. Various NLC morphology and associated gravity waves are discussed in the paper, including ice voids, a medium scale GW.

Overall, the paper is clearly written and the results are interesting. On the other hand, the English language needs to be tightened up. Please see the minor comments. Most importantly, the paper "oversells" itself. The significance is exaggerated. Except the factor of weather and doubled field of view, the balloon photographing of NLCs does not demonstrate a substantial difference from the ground photography. The PMC-Turbo experiment targets high resolution, unlike this experiment. Also unlike the authors claim, the spatial coverage of this experiment is not comparable to satellite observations. Please correct these statements.

Detailed comments:

1. title: "large-scale mesospheric dynamics". Not really, the images shown cannot be used to study large scale waves, such as tides or planetary waves.

2. line 19: "100-1450 km"?

3. line 24: "unique", well, as shown in the next sentence, the field of view of a balloon borne camera is only twice of a ground observation (1450*750 km vs. 800*550 km).

4. line 28: "is confirmed" —> "was confirmed"

5. line 30: "vertical amplitude". This is an inaccurate expression. Please change.

6. line 33: "various distances"—> "various scales"

7. line 42: "30-100 nm"

8. line 47: "NLC" —> "NLCs", and throughout the paper

9. line 50: "10000 km". This is not accurate. At high latitudes, the scale of tides and PWs are much shorter than 10000 km.

10. line 57: spell out these acronyms (AIM, SBUV)

11. line 69-71: the meaning is not clear. Please elaborate.

12. line 84: "large scale dynamics", we usually call large scale dynamics for much larger scales than 100 km. "opens new horizons", it is an improvement compared to ground observations. But this statement oversells.

13. line 103 "build" –> "built"

14. line 122: "preliminary" —> "preliminarily"

15. line 131: how about local time?

16. line 166: is comparable to PMC observations"...This is a wrong statement. The space observation is much wider than the balloon one. Figure 3 clearly shows this.

17. line 238: please define vertical amplitude. "vertical displacement"?

18. line 245-250: How's the amplitude of vertical displacement leading to more knowledge of kinetic wave energy? I also found this part exaggerated.

19. Conclusions. This is a good experiment. But none of the conclusions a-g clearly demonstrate its advantage to ground and space observations.

13.

---

## Referee Comment (RC2) · Anonymous Referee #2 · 29 Sep 2019

Review of "Stratospheric observations of noctilucent clouds: a new approach in studying large-scale mesospheric dynamics" by Dalin et al.

General comments The paper presents a new data set, namely Stratospheric balloon measurements of Noctilucent clouds. Stratospheric balloon measurements are a rather new way of observing these clouds. The author claims that this new method has several advantages as listed in the manuscript, and as such I would expect it to yield new information about the clouds. However, in my view the manuscript mostly describes the observations, and is lacking such information. I believe more insights can be drawn from the data set by performing detailed studies such as time evolution of the cloud

movements/growth. I also have an issue with the nomenclature of the paper. I do not believe these voids are the same as the much larger ice voids seen by CIPS Bailey et al. (2009) and Thurairajah et al. (2013b) or by Megner et al. (2018), since the sightings there were of a single void in an otherwise fairly homogeneous PSC cover. The formations here seem less round, and more like openings in an inhomogeneous cloud cover. Moreover, as the authors point out these features move with the wind, was not the case in Megner et al (2018). I therefore think it would cause confusion to name these by the same term 'ice voids', and suggest that the authors come up with another term.

Specific comments 153-154 and Figure 3: It is unclear if the marked area is the actual PSC coverage or the observation area, or both, i.e. that the whole observation area was filled with clouds. 189-193: Please explain how a large-scale gravity wave could produce the shapes/patterns of the cloud openings. 229-232: I do not believe one can determine that this is a vertical modulation from only one picture. If you have several pictures from different angles then this could be determined, but from one picture it is not possible to separate horizontal and vertical modulation. Figure 4 and 5: It is very difficult to judge the shape and clarity of the cloud openings when there are pre-drawn red lines to guide the eye. Please remove those. They could be replaced by arrows if you find it necessary. Technical Corrections 177: remove 'there should be fulfill'

---

## Author Comment (AC1) · 9 Oct 2019

We thank anonymous Referee 1 for her/his useful comments and suggestions, which led to improvements of our manuscript. In the revised version of the manuscript we have carefully addressed Referee 1' comments and suggestions which are highlighted in green. Our detailed replies are provided below.

Please note that the title of the manuscript has been slightly changed to "Stratospheric observations of noctilucent clouds: a new approach in studying middle- and large-scale mesospheric dynamics".

[Figure]

Anonymous Referee #1

Comment: This paper reports the stratospheric observations of noctilucent clouds from a balloon on 5-6 July 2018 near Moscow. Several hundreds of NLC images were taken for over one hour. Various NLC morphology and associated gravity waves are discussed in the paper, including ice voids, a medium scale GW. Overall, the paper is clearly written and the results are interesting. On the other hand, the English language needs to be tightened up. Please see the minor comments. Most importantly, the paper "oversells" itself. The significance is exaggerated. Except the factor of weather and doubled field of view, the balloon photographing of NLCs does not demonstrate a substantial difference from the ground photography. The PMC-Turbo experiment targets high resolution, unlike this experiment. Also unlike the authors claim, the spatial coverage of this experiment is not comparable to satellite observations.

Reply: As far as the PMC-Turbo experiment is concerned, we can note the following. This technique to image the PMC layer at scales from 100 km down to 1 m can be readily performed at the ground by using simple commercial digital cameras and tele-centric lenses. Many images similar to PMC-Turbo ones have already been obtained from the ground. One can also use a small school telescope and camera attached to it to view an NLC layer at 1 cm spatial resolution and taking several images per second, if it is necessary. Nowadays it is very easy to obtain images similar to PMC-Turbo ones from the ground, and at much lower cost compared to any balloon-borne experiment. The advantages of using balloon-borne observations of NLCs are given below in the reply as well as in the conclusions of the present paper.

Comment:

Please correct these statements.

Detailed comments: 1. title: "large-scale mesospheric dynamics". Not really, the images shown cannot be used to study large scale waves, such as tides or planetary waves.

Reply: We have corrected the title of the paper which now includes "middle-scales" since we observe both middle-scales (100-1000 km) and large-scales (1000-1500 km). Utilizing future long duration balloon flights (several days) one can study both tides and planetary waves. Since we discuss our future long duration flights as well, the title of the paper corresponds to the current and future state of the project.

Comment: 2. line 19: "100-1450 km"?

Reply: These numbers are correct and kept unchanged.

Comment: 3. line 24: "unique", well, as shown in the next sentence, the field of view of a balloon borne camera is only twice of a ground observation (1450*750 km vs. 800*550 km).

Reply: Yes, the field of view of the present experiment was twice of the ground observation. We believe it is certainly enough to call this experiment "unique". We should also note that a field of view as viewed from a balloon can be significantly extended by using multiple cameras with wide-angle lenses (a project we are currently implementing) as well as observing NLC from higher altitudes (30-38 km). The present experiment was the first in this field and it will be upgraded in the near future. We keep the definition "unique" unchanged.

Comment: 4. line 28: "is confirmed" -> "was confirmed"

Reply: It has been corrected in the revised manuscript on line 28: "which was confirmed by satellite measurements."

Comment: 5. line 30: "vertical amplitude". This is an inaccurate expression. Please change.

Reply: It has been corrected as "amplitude" on line 31 in the revised manuscript.

Comment: 6. line 33: "various distances"-> "various scales"

Reply: It has been corrected as "various scales" on line 33 in the revised manuscript.

Comment: 7. line 42: "30-100 nm"

Reply: These values are correct according to numerous measurements and model results. These values are kept unchanged in the revised manuscript.

Comment: 8. line 47: "NLC" -> "NLCs", and throughout the paper

Reply: It has been changed throughout the revised manuscript.

Comment: 9. line 50: "10000 km". This is not accurate. At high latitudes, the scale of tides and PWs are much shorter than 10000 km.

Reply: We consider mostly subpolar latitudes (59-65° N) in the present study. The distance around a latitude circle at 63° is about 18000 km. The 24 h tide and PWs with the zonal number of one have horizontal wavelengths (scales) of about 18000 km. The 12 hours tide and PWs with the zonal number of 2 have wavelengths of about 9000 km. Thus, we do not see any contradiction to the scale of 10000 km and we keep it unchanged.

Comment: 10. line 57: spell out these acronyms (AIM, SBUV)

Reply: It has been corrected on lines 59-60 in the revised manuscript: "The Aeronomy of the Ice in the Mesosphere (AIM), Odin, Solar Backscatter Ultraviolet Radiometer (SBUV) instruments)."

Comment: 11. line 69-71: the meaning is not clear. Please elaborate.

Reply: We have rephrased this meaning on lines 75-77 in the revised manuscript as follows: "At the same time, observations made from stratospheric altitudes (20-40 km) are potentially available for comprehensive studies of NLCs/PMCs."

Comment: 12. line 84: "large scale dynamics", we usually call large scale dynamics for much larger scales than 100 km. "opens new horizons", it is an improvement compared to ground observations. But this statement oversells.
Reply: We have added the information on middle- and large-scale dynamics on lines 89 and 91 since we consider both middle-scales (100-1000 km) and large-scales (1000-1500 km) in the present study as follows: "In this paper, we report on scientific results of a new balloon-borne experiment dedicated to studies of NLC middle- and large-scale dynamics at horizontal scales of more than 100 km (Dalin et al., 2019). Such experiment, conducted for the first time, opens new horizons for studies of middle- and large-scale dynamical features in combination with a high spatial resolution at the summer mesopause..."

We believe that the definition "opens new horizons" is valid for stratospheric observations of NLCs for 24 h and at large scales in case of a long duration balloon flight (several days) as we discuss in the present paper. We keep this definition unchanged in the revised manuscript.

Comment: 13. line 103 "build" -> "built"

Reply: It has been corrected on line 108 in the revised manuscript.

Comment: 14. line 122: "preliminary" -> "preliminarily"

Reply: It has been corrected on line 127 in the revised manuscript.

Comment: 15. line 131: how about local time?

Reply: The local time has been added in the revised manuscript (line 136): "a) NLCs were observed between 20:30 and 23:15 UT (23:30 and 02:15 LT) on 5 July 2018."

Comment: 16. line 166: is comparable to PMC observations"...This is a wrong statement. The space observation is much wider than the balloon one. Figure 3 clearly shows this.

Reply: Here we compare balloon-borne scales to scales of the PMC observation scene which has dimensions of 120°x80°, as measured from the nadir direction. This results in spatial coverage of approximately 2000 km along the AIM satellite track and 1000

km across track (Rusch et al., 2009). Thus, this statement is not wrong. We have added information in the revised paper (lines 174-175) as follows: "Thus, balloon-borne NLC observations have an obvious great advantage over ground-based observations in terms of larger geographic coverage which is comparable to PMC observations made from space since a PMC observation scene has spatial coverage of about 2000 km along the AIM satellite track and 1000 km across track (Rusch et al., 2009)."

Comment: 17. line 238: please define vertical amplitude. "vertical displacement"?

Reply: We define this amplitude as a semi-amplitude of a monochromatic wave, which is half of the peak-to-peak wave amplitude between highest and lowest amplitude displacement values. The definition "vertical displacement" is incorrect in this case since it ranges from zero to the amplitude of a monochromatic wave. We have added this definition in the revised manuscript (lines 247-249) as follows: "We have carefully estimated parameters of this particular wave: its horizontal wavelength was equal to $49.4 \pm 2.2$ km and its vertical amplitude was $1.9 \pm 0.1$ km between the crest and trough. We define this amplitude as a semi-amplitude A of a monochromatic wave with oscillation frequency $\omega$, which is half of the peak-to-peak wave amplitude between the highest (crest) and lowest (trough) displacement values."

Comment: 18. line 245-250: How's the amplitude of vertical displacement leading to more knowledge of kinetic wave energy? I also found this part exaggerated.

Reply: Kinetic wave energy and wave amplitude are linked by the follow relation: $E \sim 0.5*A^2*omega^2$

We have added this information in the revised manuscript (line 264) as follows: "Since wave amplitude represents kinetic wave energy ($E \sim 0.5*A^2*omega^2$), ..."

Comment: 19. Conclusions. This is a good experiment. But none of the conclusions a-g clearly demonstrate its advantage to ground and space observations.

Reply: We disagree with this comment. The conclusions a), c), f) and g) clearly demonstrate the advantages of a balloon-borne observation compared to a ground one. The conclusion e) clearly demonstrates its advantage to a space one. But it is important to realize that the combination of these points (24 h NLC observations both at middle/large and small scales) can only be achieved by using a balloon-borne observation. We have emphasized this combination at the beginning of the conclusions. We keep these conclusions unchanged in the revised manuscript.

---

## Author Comment (AC2) · 10 Oct 2019

We thank anonymous Referee 2 for her/his useful comments and suggestions, which led to improvements of our manuscript. In the revised version of the manuscript we have carefully addressed Referee 1' comments and suggestions which are highlighted in green. Our detailed replies are provided below.

Please note that the title of the manuscript has been slightly changed to "Stratospheric observations of noctilucent clouds: a new approach in studying middle- and large-scale mesospheric dynamics".

[Figure]

General comments:

The paper presents a new data set, namely Stratospheric balloon measurements of Noctilucent clouds. Stratospheric balloon measurements are a rather new way of observing these clouds. The author claims that this new method has several advantages as listed in the manuscript, and as such I would expect it to yield new information about the clouds. However, in my view the manuscript mostly describes the observations, and is lacking such information. I believe more insights can be drawn from the data set by performing detailed studies such as time evolution of the cloud movements/growth.

Reply:

We partly agree with this comment that "more insights can be drawn from the data set by performing detailed studies such as time evolution of the cloud movements/growth."

At the same time, the present paper is a review paper on the first experiment dedicated to observations and studies of NLCs at middle- and large-scales (100-1500 km). A review paper on the PMC-Turbo experiment by Fritts et al. (2018) has been recently presented in the literature, describing various observed phenomena in PMCs without a deep analysis. The authors have stated that "An overview of the PMC Turbo experiment motivations, scientific goals, and initial results is presented here... These examples suggest clear benefits of CIPS and PMC Turbo image comparisons, and subsequent PMC Turbo papers addressing specific dynamics sequences will explore several of these cases in greater detail."

We also present a review paper on our experiment describing initial results. Detailed studies will be done in the future. We have added this information in the abstract of the revised manuscript (lines 33-35) as follows: "Here we present a review paper on our experiment describing initial results. Detailed studies on time evolution of the cloud movements will be done in the future."

Comment:

I also have an issue with the nomenclature of the paper. I do not believe these voids are the same as the much larger ice voids seen by CIPS Bailey et al. (2009) and Thurairajah et al. (2013b) or by Megner et al. (2018), since the sightings there were of a single void in an otherwise fairly homogeneous PSC cover. The formations here seem less round, and more like openings in an inhomogeneous cloud cover. Moreover, as the authors point out these features move with the wind, was not the case in Megner et al (2018). I therefore think it would cause confusion to name these by the same term 'ice voids', and suggest that the authors come up with another term.

Reply:

We believe it is unreasonable to introduce a new definition for a similar morphological phenomenon observed in NLCs/PMCs structures.

Thus, Figure 3c of Thurairajah et al. (2013a) as well as Figure 1, Figures 2a-3, 2a-4, Figure 3a, Figure 4a, Figure 5b, Figures 6a,c, Figure 7a of Thurairajah et al. (2013b) clearly demonstrate the presence of partial ice voids at the edges of the PMC area. Figure 9 of Rusch et al. (2009) also illustrates partial ice voids at the PMC edge. Figure 2 of Megner et al. (2018) shows the particular ice void having an incomplete oval shape, i.e., this was a partial ice void observed in NLCs.

We keep the definition "partial ice voids" unchanged in the revised manuscript.

Specific comments:

153-154 and Figure 3: It is unclear if the marked area is the actual PSC coverage or the observation area, or both, i.e. that the whole observation area was filled with clouds.

Reply:

The marked area is the actual NLC coverage. The observation area is larger. We have added this information on line 159 in the revised manuscript and in the caption of Figure 3: "One can see that the NLC field (their actual coverage) extended mostly from the west to east along an area filled with low temperatures of 136-146 K..."

Comment:

189-193: Please explain how a large-scale gravity wave could produce the shapes/patterns of the cloud openings.

Reply:

This has been explained a bit earlier (lines 185-186 of the previous manuscript and on lines 193-195 of the revised manuscript): "Rusch et al. (2009) have hypothesized that ice voids could be caused by heating due to the passage of warm crests of a gravity wave."

Comment:

229-232: I do not believe one can determine that this is a vertical modulation from only one picture. If you have several pictures from different angles then this could be determined, but from one picture it is not possible to separate horizontal and vertical modulation.

Reply:

We are not 100% sure we understand this comment. Wave horizontal and vertical modulations are at the right angle to each other, and thus they do not produce a mutual interference. But we did analyze nine images at various viewing angles in order to deduce the maximum vertical displacement (amplitude) of this particular wave. The nine images showing progressive changes in the wave displacement can be found at the following webpage: ftp://ftp.irf.se/outgoing/pdalin/NLC/SONC_experiment_2018_07_05/WAVE_AMPLITUDE/

We have added this information in the revised manuscript (lines 256-260): "We have analysed nine images at various viewing angles in order to deduce the maximum vertical displacement (amplitude) of this particular wave. The nine images showing progressive changes in the wave vertical displacement can be found at the following webpage: ftp://ftp.irf.se/outgoing/pdalin/NLC/SONC_experiment_2018_07_05/WAVE_AMPLITUDE/"

Comment:

Figure 4 and 5: It is very difficult to judge the shape and clarity of the cloud openings when there are pre-drawn red lines to guide the eye. Please remove those. They could be replaced by arrows if you find it necessary.

Reply:

We have replaced the red lines with arrows indicating the centers of the partial ice voids. Please see new versions of Figure 4 and 5 attached.

Comment: Technical Corrections 177: remove 'there should be fulfill'

Reply:

This has been removed.

[Figure]

NLC on 5-6 July 2018,  H=4092 m

[Figure]

H=4947 m

[Figure]

H=7836 m

[Figure]

H=9077 m

[Figure]

H=13928 m

[Figure]

**Fig. 1.**

[Figure]

[Figure]

[Figure]

**Fig. 2.**

---

## Author Response (AR2)

**Dear Editor,**
**Please consider our revised manuscript according to the comments of the**
**Reviewer 3. We thank the Reviewer 3 for her/his useful comments, which led to**
**improvements of our manuscript. In the revised version of the manuscript we**
**have carefully addressed the comments of the Reviewer 3 which are highlighted**
**in green. Our detailed replies are provided below.**

Submitted on 20 Nov 2019
Anonymous Referee #3

Suggestions for revision or reasons for rejection (will be published if the paper is
accepted for final publication)

This revised paper describes NLC observations from a stratospheric balloon carried
out during a campaign in the Moscow area during the 2018 NLC season. The
observations are novel and have the potential to provide new and interesting insight
into NLC and the processes affecting NLCs. The paper does not really present
spectacularly new results, but is still an interesting contribution to the field and should
be published subject to some minor revisions, in my opinion. The paper is overall well
written and easy to follow.

Specific comments:

Line 183 following: the vertical gradient of potential temperature was determined
from MLS temperature measurements. The vertical resolution of MLS temperature
profiles in the upper mesosphere is with 12 – 14 km quite low. Given this poor
vertical resolution, how reliable can estimates of the potential temperature gradient at
NLC altitude be? I'm not convinced one can draw robust conclusions.

**We use the newest version of the Aura/MLS temperature data (ver. 4.23).**
**According to the data quality document (available at**
**https://mls.jpl.nasa.gov/data/v4-2_data_quality_document.pdf), the vertical**
**temperature resolution is 6 km at 0.01 hPa and 8–10 km at 0.001 hPa. Since we**
**consider the temperature data in the mesosphere and up to the mesopause level**
**(0.0046 hPa), the actual vertical temperature resolution is about 6–8 km (not 12–**
**14 km). Since we have considered the temperature data covering a large**
**geographic area (the NLC area observed from the balloon) and we could not find**
**any negative values of the potential temperature gradient, we consider this**
**conclusion to be robust.**

Line 241 following: How can you distinguish between an altitude variation and a
variation of the horizontal distance between the cloud and the balloon. In other words,
how can you distinguish between a vertical and a horizontal wave pattern? I don't
think a unique altitude determination is possible based on the ballon images only.

**First. We consider relative variations of a wave motion, i.e., ±variations**
**relative to some undisturbed level. We use snapshots with a short exposure time**
**of 1/8 sec. The horizontal speed of the balloon was about 16 m/s at the time of**
**observing the wave with vertical disturbances. Thus, the horizontal motion of the**
**balloon was equal to 2 metres during a snapshot which yields negligibly small**
**uncertainties for estimating relative vertical variations of the wave motion. The**

**position of the balloon was known and was taken into account in the calculations.**

**Second. For this peculiar wave (vertical wave pattern) we assume that the observed wave disturbances occur in the vertical plane only. If a significant horizontal wave pattern were present at that time we would get different vertical amplitudes for the wave crest and trough due to different changes in the elevation angles in the wave crest and trough. But since we have estimated the same wave amplitudes both for the wave crest and trough it means that the wave motion was indeed in the vertical plane.**

**Thus a unique altitude determination based on the balloon images is possible. We have added this information on lines 247-250 in the revised manuscript.**

Line 258: "kinetic wave energy"
Strictly speaking, the „E" in the equation provided is not an energy, i.e., its unit is not Joule.
**Yes, it is correct. This is the wave energy per unit mass. We have corrected it on lines 261-262 in the revised manuscript.**

Line 531: „at the mesopause (86.1 km)"
Is this geometric or geopotential altitude?
**This is geometric altitude. We have corrected it in the caption of Figure 3.**

Typos, grammar etc.:
Line 67: "are limited to" -> "are limited by"
**It has been corrected.**

Line 174: "those centers" -> "whose centers" ?
**It has been corrected.**

Line 525: "build" -> "built"
**It has been corrected.**

Line 529: "trajectories" -> "trajectory"
**It has been corrected.**

[revised manuscript text omitted]